# A systems genomics approach uncovers molecular associates of RSV severity

**Matthew N. McCall**[1,2], **Chin-Yi Chu**[3,4☯], **Lu Wang**[1☯], **Lauren Benoodt**[5], **Juilee Thakar**[1,6], **Anthony Corbett**[1,7], **Jeanne Holden-Wiltse**[1,7], **Christopher Slaunwhite**[3,4], **Alex Grier**[6], **Steven R. Gill**[6], **Ann R. Falsey**[8,9], **David J. Topham**[6,10], **Mary T. Caserta**[4], **Edward E. Walsh**[8,9], **Xing Qiu**[1]*, **Thomas J. Mariani**[3,4]*

1 Department of Biostatistics and Computational Biology, University of Rochester Medical Center, Rochester New York, United States of America, 2 Department of Biomedical Genetics, University of Rochester Medical Center, Rochester New York, United States of America, 3 Division of Neonatology and Pediatric Molecular and Personalized Medicine Program, University of Rochester Medical Center, Rochester New York, United States of America, 4 Department of Pediatrics, University of Rochester Medical Center, Rochester New York, United States of America, 5 Department of Biochemistry and Biophysics, University of Rochester Medical Center, Rochester New York, United States of America, 6 Department of Microbiology and Immunology, University of Rochester Medical Center, Rochester New York, United States of America, 7 Clinical and Translational Science Institute, University of Rochester Medical Center, Rochester New York, United States of America, 8 Department of Medicine, University of Rochester Medical Center, Rochester New York, United States of America, 9 Department of Medicine, Rochester General Hospital, Rochester New York, United States of America, 10 David H. Smith Center for Vaccine Biology and Immunology, University of Rochester Medical Center, Rochester New York, United States of America

☯ These authors contributed equally to this work.
* xing_qiu@urmc.edu (XQ); Tom_Mariani@urmc.edu (TJM)

**Data Availability Statement:** Complete molecular and microbiota data for these studies is available in

## Abstract

Respiratory syncytial virus (RSV) infection results in millions of hospitalizations and thousands of deaths each year. Variations in the adaptive and innate immune response appear to be associated with RSV severity. To investigate the host response to RSV infection in infants, we performed a systems-level study of RSV pathophysiology, incorporating high-throughput measurements of the peripheral innate and adaptive immune systems and the airway epithelium and microbiota. We implemented a novel multi-omic data integration method based on multilayered principal component analysis, penalized regression, and feature weight back-propagation, which enabled us to identify cellular pathways associated with RSV severity. In both airway and immune cells, we found an association between RSV severity and activation of pathways controlling Th17 and acute phase response signaling, as well as inhibition of B cell receptor signaling. Dysregulation of both the humoral and mucosal response to RSV may play a critical role in determining illness severity.

## Author summary

This paper presents a novel approach to understanding the localized molecular responses to respiratory syncytial virus (RSV) and the system-level correlates of clinical outcomes. To do this, we developed a novel statistical method able to integrate high dimensional

dbGaP (phs001201.v2.p1). All code and ancillary data are available on GitHub: https://github.com/mccallm/aspires.

**Funding:** This project has been funded with Federal funds from the National Institute of Allergy and Infectious Diseases, National Institutes of Health, Department of Health and Human Services, under Contract No. HHSN272201200005C (DJT), a University of Rochester School of Medicine and Dentistry Scientific Advisory Committee Incubator grant (TJM, XQ, EEW), and University of Rochester Center for Clinical & Translational Science Institute grant number UL1 TR002001 (Zand/Bennet). The funders had no role in study design, data collection and analysis, decision to publish, or preparation of the manuscript.

**Competing interests:** I have read the journal's policy and the authors of this manuscript have the following competing interests: ARF is currently receiving funding from Merck Sharpe and Dohme, Pfizer, Janssen, Astra Zeneca and BioFire and personal fees for DSMB from Novavax. The others authors do not have any competing interests to report.

molecular data characterizing the host airway microbiota and immune and nasal gene expression. We show that this integrative approach facilitates superior performance in estimating clinical outcome as opposed to any single data type. Using this approach, we identified both cell type-specific and shared biomarkers and regulatory pathways associated with RSV severity. Specifically, we identified an association between RSV severity, activation of pathways controlling Th17, and inhibition of B cell receptor signaling, which were present in both the site of infection airway and in peripheral immune cells. These results can guide future efforts to identify biomarkers for identifying or predicting illness severity following infant RSV infection. They may also be useful as biomarkers to inform the efficacy of future interventions (e.g., therapies) or preventative measures to suppress the rate of severe disease (e.g., vaccines).

## Introduction

Respiratory syncytial virus (RSV), a negative strand RNA virus in the Pneumoviridae family, is a major cause of respiratory illness affecting persons of all ages, especially newborn infants [1–3]. Although the majority of infections are relatively mild, RSV remains the most common cause of hospitalization for pneumonia and severe pneumonia in infants and young children in both the developed and developing world [4–6]. In the US half of newborns are infected in their first winter, with 1–3% hospitalized, 4–7% seen in emergency departments, and 10–16% seen in physician offices [7].

A number of well-defined host factors predisposing infants to severe disease include prematurity, congenital cardiac and neuromuscular disease, and low levels of maternally derived neutralizing antibody [3,8]. More recent studies have also found genetic polymorphisms in cytokine and chemokine genes, altered innate interferon responses in the respiratory tract, T cell responses favoring a Th2 and Th17 bias, and the composition of the nasal microbiota to be associated with more severe illness [8–15]. Although each of these factors offer insight into the complex nature of RSV infection in young infants, they have generally been analyzed independently; thus, it is difficult to assess their interactions and relative importance to disease outcome.

Previous multi-omic analyses of RSV, by our group and others, demonstrated the potential of integrative analyses to further our understanding of the biological mechanisms underlying RSV disease progression and severity. To address limitations of prior studies, we designed a systems-level study of RSV pathophysiology in a precisely defined population of low risk newborns with the full spectrum of disease severity [16]. We studied purified populations of CD4+, CD8+ and CD19+ cells, as mixed PBMCs have been used extensively in studies of illness severity for respiratory infections [9,11,17–19]. We reasoned that the local airway response would be a key component to defining illness severity, by contributing to the ability of the host to control or localize the infection [20]. We also included data from the nasal microbiota, as recent studies have indicated colonization at the time of viral infection may significantly influence illness severity [9].

This study builds on these previous studies both in the scope of the data and in the methodology developed to analyze these data. By modeling the connections between these high-throughput data and clinical RSV severity, we are able to reconstruct the intricate relationships among different data types and demonstrate the potential of integrative analyses to identify shared and cell type specific cellular pathways associated with RSV severity.

## Results

The data presented in this manuscript were generated as part of the Assessing Prediction of Infant Respiratory Syncytial Virus Effects and Severity (AsPIRES) study, which sought to identify host, viral, and environmental factors associated with RSV disease severity [21]. A total of 139 infants with RT-PCR confirmed RSV illness were enrolled, of which 134 had omics data of sufficient quality for analysis (demographic information for this cohort is shown in Table A in S1 Text). Venous blood, nasal microbiota and nasal epithelial cell samples were collected for high-throughput molecular analysis (Fig 1). Illness severity was measured using the Global Respiratory Severity Score (GRSS) [22], which quantifies the full spectrum of primary RSV disease severity using nine clinical variables in a weighted score. We employ a novel approach to the integration and analysis of five high dimensional omic data modalities: the nasal epithelial transcriptome, the transcriptome of CD4, CD8, and CD19 cells from peripheral blood, and the nasal microbiome.

### Integrated method development

Preliminary exploratory analyses of individual data types found that most individual features (e.g. a single gene measured in CD4 cells) have relatively weak correlation with GRSS (median absolute correlation 0.08–0.11). Additionally, we observed strong correlation among features within a data type, which are typical of transcriptomic data. These two observations motivated us to use Principal Components Analysis (PCA) to aggregate numerous "weak features" in the hopes of identifying a few key latent factors for each data type. Screeplots from performing PCA on each data type showed that a small number of Principal Components (PCs) explain the vast majority of the variation in each data type (S1 Fig). This further supports our observation of strong correlation among features within each data type. Furthermore, we applied a secondary PCA on the PCs produced by each data type and found shared variation between data types. Specifically, secondary PCs contained primary PCs from different data types, and

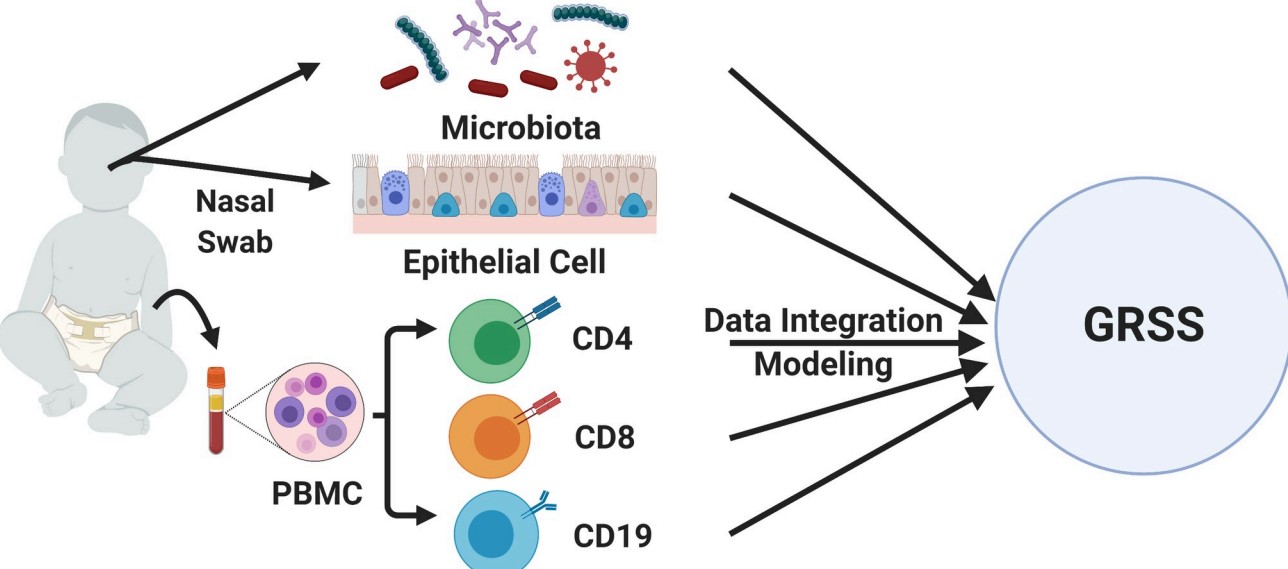

**Fig 1. Overview of the study design.** Measurements of the nasal epithelial cell transcriptome and nasal microbiome were generated from nasal swabs of infants. Measurements of peripheral immune cells (CD4, CD8, and CD19) were obtained from blood samples. These measurements were subsequently integrated and associated with RSV disease severity (GRSS).

the number of secondary PCs was far less than the number of primary PCs. These initial analyses motivated the methods of data integration we describe in this paper.

Due to practical limitations, only a small subset of subjects (23 out of 106) have all five high dimensional data types (S2 Fig). Consequently, we decided to conduct integrative analyses of several combinations of omics data. The first set of data modalities was chosen to interrogate three disparate putative determinants of RSV severity, the nasal epithelial transcriptome (NT), the nasal microbiome (NM), and the adaptive immune response measured in peripheral blood CD4 cells (CD4). These data were available for 61 subjects. The second data integration focused on the collective role of the adaptive and innate immune response in RSV severity, measured in CD4, CD8, and CD19 cells isolated from peripheral blood. These data were available for 35 subjects. Demographic information for these sub-cohorts is shown in Tables B & C in S1 Text.

We proposed five related integrative analytic methods based on multilayer PCA and regularized regression based variable selection (Fig 2), see Materials and Methods for details. Based on extensive cross-validation (CV) experiments, we found that PCA of transcriptomic data, followed by an integrative elastic-net regression model with the transcriptomic PCs and individual operational taxonomic units (OTUs) of nasal microbiome data (Fig 2, Method 1) achieved better GRSS prediction accuracy than the other methods (Table D in S1 Text).

We also applied a similar approach, initial PCA-based dimension reduction followed by regularized regression, to individual data types and found that the integrative models significantly out-performed the single data type models in terms of cross-validated prediction accuracy (Fig 3). Specifically, while the integrated and single data type models are all approximately unbiased, the integrated models had substantially smaller mean squared error (MSE).

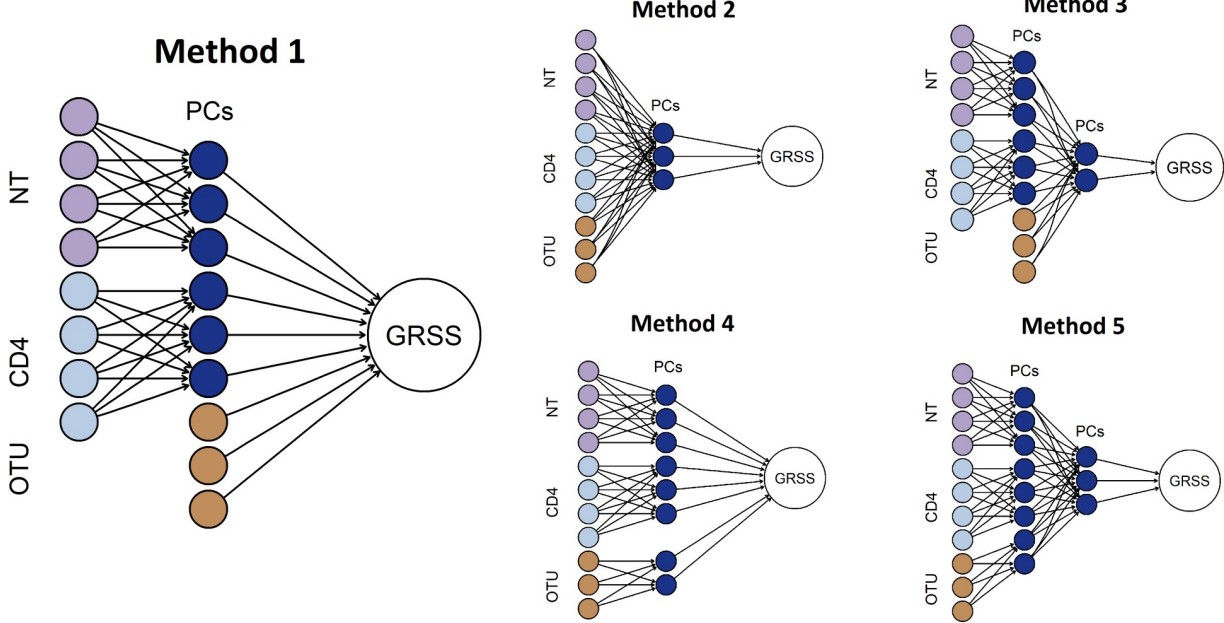

**Fig 2. Five potential methods of multi-omic data integration.** Methods differ in their handling of nasal microbiome data (OTU), number of levels of PCA, and the stage at which the integration occurs. Here light purple circles represent nasal gene expression profiles (NT), light blue ones represent CD4 gene expressions, and gold ones represent nasal microbiome abundance data (OTU). Dark blue ones are principal components (PCs) computed from the original features. Large white circle represents the GRSS, which is the clinical variable of interest of this study. In our assessments, the leftmost model out-performed the other models in terms of cross-validated error.

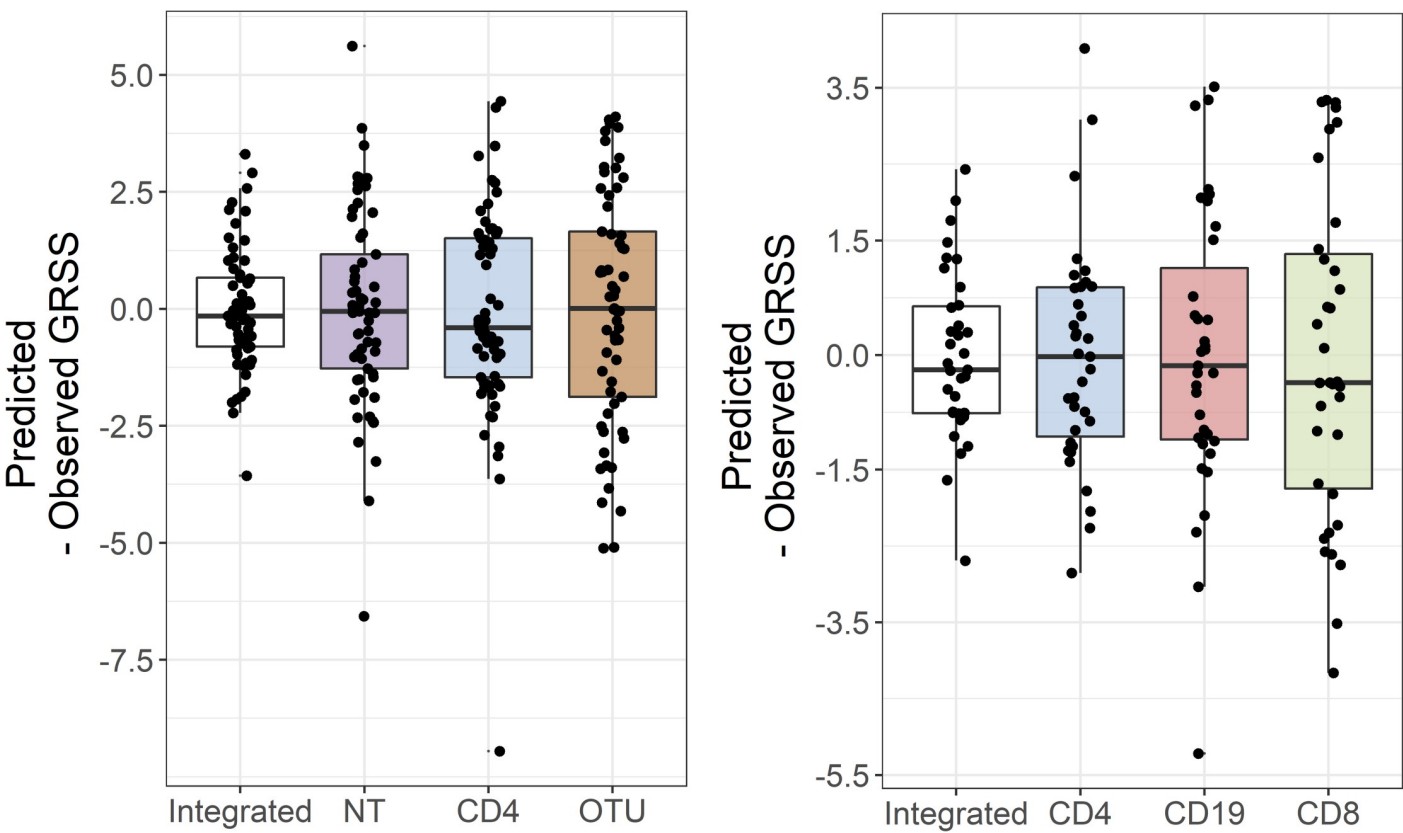

**Fig 3. Integration of different data modalities improves prediction of GRSS.** The panel on the left shows the difference between the predicted and observed GRSS for a model that integrates the nasal epithelial transcriptome, peripheral blood CD4 transcriptome, and nasal microbiome, as well as for models using just one of these data types. Similarly, the panel on the right compares a model that integrates the transcriptomes of 3 immune cell types measured in peripheral blood with models using just one of these data types. In both cases, integration increases the precision of the GRSS predictions.

## Model application and interpretation

We previously reported gene expression correlates of clinical disease severity in RSV infected infants [11,23]. To gain further insight, we generated multiple "models" by integrating unbiased, comprehensive gene expression data from both the humoral and mucosal compartments as described above. We hypothesized that a "systems level" analytical approach would provide distinct biological insights into disease pathophysiology. We first focused upon humoral responses specifically characterized in CD4 T cells sorted from peripheral blood collected during acute illness, in two models containing gene expression data from these cells (Fig 4). Following data integration, using RSV-associated disease severity as the outcome, the modeled weights (see "Feature weight calculation" in S1 Text for more details) for expression of individual genes is displayed in word clouds (upper left), and unweighted gene expression values are displayed in a heatmap (lower left). Weights for individual genes are clearly different between the two models, as evident from the word clouds, and could be expected due to the different subsets of the cohort included in each model. Interestingly, CD101, one of the highest weighted (absolute value 0.0048) genes in both models, plays a role as an inhibitor of T cell proliferation induced by CD3. Furthermore, unweighted gene expression in these integrated models are not fundamentally different between mild and severe subjects. These observations support the novel and distinct insight derived from our new integrated modeling approach to identify gene expression correlates of disease severity.

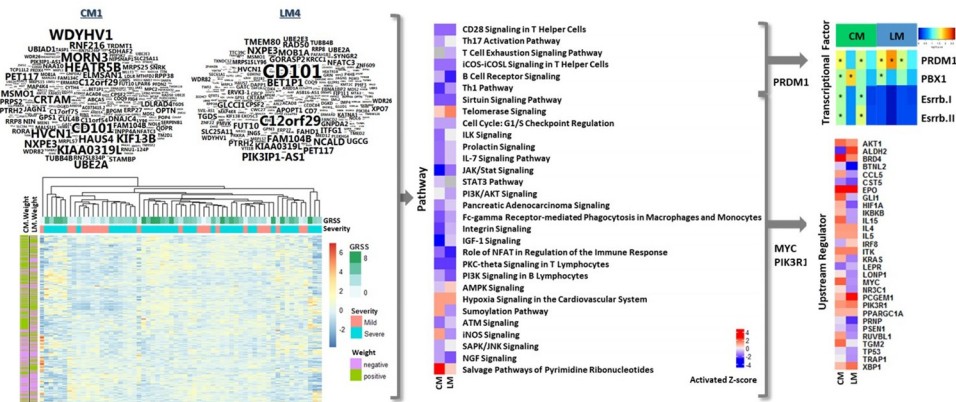

**Fig 4. CD4 gene weights, expression, pathways, transcriptional factors and upstream regulators associated with clinical severity in integration models.** Shown are an integration model of CD4, nasal epithelial cells and microbiota (CM) and a model of lymphocytes (LM). Weights generated by integration models are shown in word-clouds. The size of word represents the absolute value of gene weight. Word-clouds of CM & LM consist of genes that have absolute weight is greater than 0.0003. Gene expression are normalized expression levels for the 454 genes selected by univariate analysis; rows represent genes and columns represent samples. Red indicates higher expression, blue indicates low/no expression, green indicates Global RSV Severity Score (GRSS), soft orange indicates mild phenotype, lime green indicates severe phenotype, purple indicates negative weight and olive indicate positive weight. Transcriptional factors associated with severity in CD4 lymphocytes were identified using a hypergeometric test. Four transcriptional factors are shown where p-values were less than 0.05. Ingenuity Pathway Analysis (IPA) was used to identify canonical pathways and upstream regulators represented by genes associated with severity in CD4 lymphocytes. Thirty pathways and upstream regulators are shown where Fisher's exact test p-values were less than 0.05. Red and blue indicate predicted increased or decreased pathway activation (activation z-score), respectively.

To minimize model-specific variation, we focused our interpretation of these data at the pathway level (middle of Figs 4 and S3–S12). We found a high degree of model convergence, with a majority (95%) of pathways consistently, significantly activated or inhibited (p-value < 0.05) in association with disease severity. Significantly activated pathways were associated with hypoxia signaling, nucleotide salvage and telomerase signaling pathways. Of particular interest, many significantly inhibited (p-value < 0.05) pathways were associated with T helper (Th) cell signaling (CD28 and NFAT signaling) and differentiation of Th subtypes (Th1 and TH17 regulation).

We next attempted to predict key regulatory events associated with disease severity, including transcription factors and intracellular signal transduction molecules, that could be driving the global gene expression responses. Upstream regulator analysis using Ingenuity suggested that MYC and PI3K were associated with many of the significantly regulated (p-value < 0.05) pathways (Fig 4). Four sets of genes were selected based on their model weights: the top 200 genes, genes in the first quantile, genes in the second quantile, and all genes. Promoter analysis of these three sets were performed, using high quality transcription factor binding sites conserved across human, mice and rat genomes. The analysis revealed that multiple T cell differentiation pathways with consistent evidence for inhibition (CD28 signaling, Th17 activation, Th1 and T cell exhaustion signaling) are under the transcriptional control of PRDM1 (Fig 4 upper right). Interestingly, PRDM1 is known to modulate peripheral T cell activation and proliferation, promote T helper (Th2) lineage commitment and limit Th1/Th17/Tfh cell differentiation. Our results implicate inhibition of T cell differentiation, towards the Th2/Th17 phenotype in particular, as a putative mediator of severe illness.

We rationalized that the mucosal response would be distinct from the humoral response and would reflect the pathophysiology of the disease target organ, as suggested by our prior

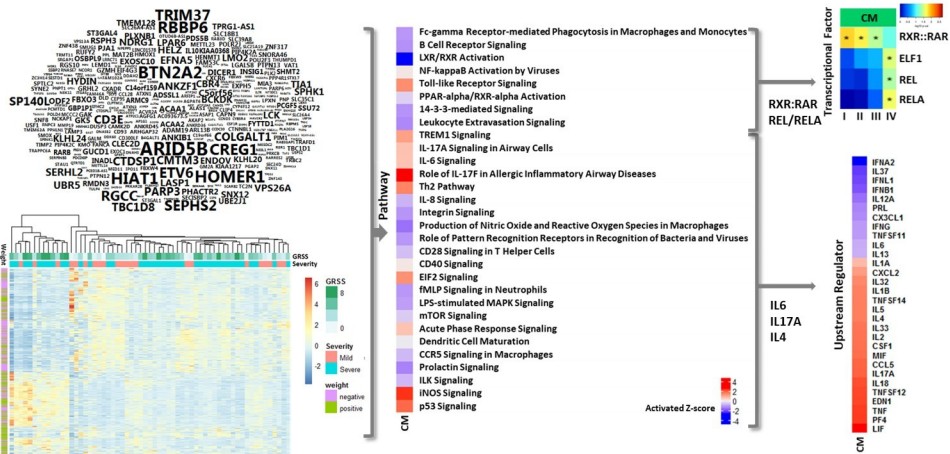

**Fig 5. Nasal Epithelial gene weights, expression, pathways, transcriptional factors and upstream regulators associated with clinical severity in integration models.** Shown are Integration model of CD4, nasal epithelial cells and microbiota (CM). Weights generated by integration models are shown in world-clouds. The size of word represents the absolute value of gene weight. A M1 word-cloud consists of genes with absolute weight greater than 0.0003. Gene expression are normalized expression levels for the 993 genes selected by univariate analysis; rows represent genes and columns represent samples. Red indicates higher expression, blue indicates low/no expression, green indicates Global RSV Severity Score (GRSS), soft orange indicates mild phenotype, lime green indicates severe phenotype, purple indicates negative weight and olive indicate positive weight. Transcriptional factors associated with severity in nasal epithelial cells were identified using a hypergeometric test. Four transcriptional factors are shown where p-values were less than 0.05. Ingenuity Pathway Analysis (IPA) was used to identify canonical pathways and upstream regulators represented by genes associated with severity in nasal epithelial cells. Thirty pathways and upstream regulators are shown where Fisher's exact test p-values were less than 0.05. Red and blue indicate predicted increased or decreased pathway activation (activation z-score), respectively.

studies [11,24]. Therefore, we next focused on interpreting the gene weights of nasal samples from the comprehensive model (CM). Similar to the CD4 data presented in Fig 4, the modeled weights for expression of individual genes in relation to disease severity is displayed in word clouds (Fig 5, upper left), and actual unweighted gene expression values are displayed in a heatmap (Fig 5, lower left). Highly weighted genes (absolute value greater than 0.0003) that were positively correlated with disease severity include BTN2A2, which inhibits the proliferation of CD4 and CD8 T-cells, T-cell metabolism and IFN secretion, and HOMER1, which negatively regulates T cell activation by inhibiting NFAT pathway.

We next completed pathway level analysis of the comprehensive model (CM) nasal gene expression weights (Figs 5, center and S3–S12). The analysis uncovered a decrease in multiple pathways driven by both retinoic acid-related (e.g., LXR/RXR and PPAR/RXR signaling) and p53-related (e.g., NF-kappaB) signaling were associated with severe disease. Further analysis of these genes and pathways identified significant evidence (p-value < 0.05) for changes in RXR and REL/A transcription factor regulation (Fig 5, upper right). Interestingly, this analysis indicated increased activation of pathways which were focused on regulation of the immune system were also associated with severe disease; in particular those associated with Th2 and Th17 CD4 T cells. Remarkably, upstream regulator analysis suggested this may be due to increases in the expression of IL4 and IL17A (Fig 5, lower right). Again, as for CD4 T cell data, our ability to use this integrated modeling approach to identify evidence for pathophysiologically relevant interactions between the mucosal and humoral responses supports its methodological validity.

Finally, we assessed unique and consistent responses in the mucosal and humoral compartments as indicated by our integrated models (CM and LM). Pathway-based analysis of the

weights derived from CD8 T cells indicated unique activation of cytotoxic responses including those related to classical CD8 T cell functions were associated with severe disease. CD8 T cells also demonstrated unique activation of TGFb and TNFR signaling in severe disease. Conversely, analysis of the weights derived from CD19 B cells indicated regulation of multiple, alternate pathways. CD19 B cells displayed evidence for unique activation of PLC, and unique inhibition of PI3K/AKT signaling, among others. Pathway-based analysis of the weights also identified a number of responses that were conserved across all lymphocytes (e.g., CD4, CD8, CD19) and associated with disease severity. Consistently activated pathways indicated broad increases in oxidative phosphorylation, nucleotide salvage and sumoylation. Significantly inhibited pathways indicated broad reductions in lymphocyte proliferation, activation (e.g., iCOS and NFAT) and surtuin signaling. Finally, we looked for pathways which were consistently identified not just in lymphocytes, but across all humoral and mucosal data sets ([Fig 6]). This analysis indicated activation of pathways controlling Th17 and acute phase response signaling, as well as consistent inhibition of B cell receptor signaling, are consistently associated with disease severity in all cell types studied.

Two alternative integration models were also considered in our study: one integrating CD4 and CD19 lymphocytes (S13–S16 Figs) and another integrating nasal epithelial cells with CD4 and CD19 lymphocytes (S17–S22 Figs). These additional models produced similar pathway level results for each transcriptomic data type, suggesting a degree of robustness to our approach to data integration and analysis and support our decision to focus on a comprehensive model of nasal epithelial cells, nasal microbiota, and CD4 lymphocytes (CM) and a model focused on three lymphocyte cell types (LM).

## Discussion

In summary, we conducted a multi-omics study on infant RSV infection. We demonstrated that a multi-layer statistical learning framework was better at predicting disease severity than

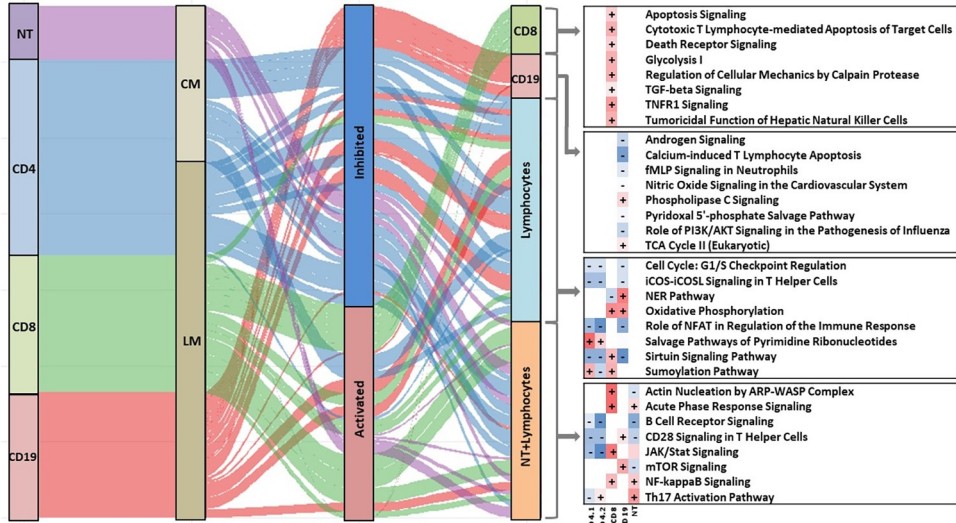

**Fig 6. Common and unique pathways in lymphocytes & nasal epithelial cells of integration models.** Sankey diagram showing the common and unique pathways among lymphocytes & nasal epithelial cells of integration model of CD4, nasal epithelial cells and microbiota (M1) and model of lymphocytes (M4). Ingenuity Pathway Analysis (IPA) was used to identify canonical pathways represented by genes associated with severity in lymphocytes and nasal epithelial cells. Pathways are shown where Fisher's exact test p-values were less than 0.05. Red and blue indicate predicted increased or decreased pathway activation (activation z-score), respectively. The width of the flow bar is proportional to absolute value of activation z-score.

comparable single-layer approaches; and integrating multiple omics datasets provided with us better prediction accuracy of disease severity than predictors built from any single dataset. In addition, based on the trained sparse linear predictors, we were able to assess the contribution of individual genes/microbes by quantitative weights, which facilitate the biological interpretations of these predictive models.

Previous studies, including our own, utilized transcriptomic analysis of single sample types such as whole blood, purified T cells, or nasal secretions to investigate RSV disease pathogenesis. These analyses found an association between increased disease severity and both T and B cell suppression, evidence of systemic Th2 skewed T cell responses, alterations in systemic and local interferon responses, and the potential influence of the local microbiome on these responses. In the current work, by integrating gene expression data from multiple cell types from the peripheral blood and the respiratory epithelium with the complex microbiome of the upper respiratory tract, we confirmed the deregulation of the immune profile associated with RSV disease severity. Specifically, the results demonstrated that Th2 and Th17 activation, and inhibition of Th1 pathways dominate the T cell response. In addition, there was evidence of B cell suppression in the airway of infants with severe RSV. The results also demonstrated that inclusion of the microbiome, specifically *Haemophilus influenzae*, was informative for understanding a complete picture of RSV disease pathogenesis in young infants. Because our microbiome analysis did not identify *Streptococcus pneumoniae*, we were unable to confirm its influence. As described in the results, our analysis identified many severity-associated pathways that were activated or suppressed during infection, especially those indicating immune suppression. As shown in Fig 5, the airway cells contributed importantly to this inhibition. Airway epithelial cell expression identified BTN2A2 as an influential gene and was important in inhibiting CD4 and CD8 cells and interferon suppression.

There are several notable limitations to this study. First, the sample size was limited due to the complex nature of the study, challenges enrolling infants with RSV, and the need for multiple visits to obtain samples. In the pathway and transcription factors analyses, we assessed statistical significance using a p-value threshold of 0.05 without adjustment for multiple testing; therefore, these results should not be viewed as definitive. The same factors that contributed to the small sample size also contributed to a second limitation: the potentially non-random missingness of certain omic data types for some participants. However, we believe that this latter limitation represents an opportunity for future methods development in multi-omic data integration. Second, a common limitation of clinical genomic studies is that sample are collected over time, which has the potential to introduce technical variation. Third, samples were collected at specific time points; therefore, it is possible that some changes in cellular function that were not temporally proximal were not captured.

Due to the high-throughput nature of omics data, there could be thousands of features that are potentially associative with biological endpoints. Besides, there exist an astronomically large number of possible interdependences among those omics data. Consequently, we believe that the most fundamental challenge in multi-omics studies such as ours is to reduce the complexity in statistical models in a sensible way, such that most spurious correlations are removed yet the major modes of informative relationships are retained.

Last but not the least, we believe our proposed method has better interpretability than those machine learning algorithms based on deep neural networks. In fact, the two methodologies share one architectural similarity, namely, both are multi-layered feedforward systems (see Fig 2). Due to the deliberate choice of using linear activation function (via PCA) and linear output function (via glmnet regression), we were able to design a multilayer backpropagation algorithm that translates the weights at the output layer, i.e., linear coefficients in the trained predictor of GRSS, to mathematically equivalent weights at the input layer, i.e., individual genes

and microbes. In a sense, the absolute value of a particular weight represents the contribution of this gene/microbe in predicting the GRSS. We performed gene set enrichment analyses based on these weights and discovered both cell-type specific and shared severity-associated biological pathways. In contrast, it is generally not possible to obtain such a direct relationship between the output and input layer in feedforward neural networks with nonlinear activation functions, which is why these algorithms are sometimes referred to as "blackbox methods".

Arguably, several components in our integrative analysis pipeline may be replaced by other more specialized methods. For example, instead of using standard PCA, we could consider probabilistic PCA [25] or sparse PCA [26]. The former is more robust to outliers and missing values due to the use of a ridge-like regularization term; and the latter is not only more consistent for "large p, small n" data than the standard PCA, it also has better interpretability because it can produce sparse loading scores. In addition, there is a host of recently developed variants of PCA that are advantageous in various situations, as summarized in [27].

Notwithstanding their advantages, these advanced dimension reduction methods are: (a) computationally more demanding; and (b) dependent on the tuning parameter that may have to be selected by cross-validation, which adds more computing cost and uncertainties in the analyses. In the future, we plan to systematically evaluate the impact of different dimension reduction strategies to the second stage integrative analyses and make algorithmic adaptations to improve their computational efficiency in high-throughput data analysis if necessary.

Another potentially rewarding future research direction is to design better statistical methods optimized for "incompletely matched" data like ours, in which all subjects do not have all types of data. In fact, as seen in S2 Fig, only 23 subjects have all five types of high-throughput data, thus we decided to build integrative models for several combinations of data separately. Development of efficient and unbiased high-dimensional imputation methods in the future may allow us to fully integrate all available data and improve the accuracy of the predictive models.

What we present in this paper is a holistic approach to understanding the response to RSV infection during infancy, and the system-level correlates of clinical outcomes. The data provide a better resolution for accurately and sensitively identifying molecular changes associated with illness severity, and also uncover specific and robust changes that may be easily detectable using distinct biospecimens (e.g., airway, T cells, B cells, etc.). These novel data can guide future efforts to identify sensitive and specific biomarkers for identifying or predicting outcome following infant RSV infection. They may also be useful as biomarkers to inform the efficacy of future interventions (e.g., therapies) or preventative measures to suppress the rate of severe disease (e.g., vaccines). For example, our approach could potentially be used to quantify the response to novel RSV vaccines, either live attenuated or subunit vaccines, to be certain they do not mimic responses that may lead to a pathologic state [28,29].

## Materials and methods

### Ethics statement

The Research Subject Review Board of the University of Rochester and Rochester General Hospital approved the study (IRB#42632), and all parents provided written informed consent.

### Study population and sample collection

The full description of the AsPIRES study has been published [21]. The Research Subject Review Board approved the study and all families provided informed consent. Briefly, three groups of previously healthy term infants with primary RSV infection were evaluated during three winter seasons from 2012 to 2015 in Rochester, NY. RSV infection was identified in

hospitalized infants, outpatients brought to medical attention either at the Emergency Department or primary care offices for respiratory symptoms, and infants in a birth cohort followed prospectively in their homes for RSV infection. From these three groups, infants with a range of RSV disease severity were included.

Infants were evaluated at three time points: an acute illness visit at diagnosis, a second visit ~14 days after illness onset, and a convalescent visit 28 days after illness onset. At each visit symptoms were recorded, and a physical exam was performed. At the first and third visit a flocked swab (Copan) was used to obtain a nasal specimen from one nares nares for microbiological testing, a nasal wash was performed on the contralateral nares to remove mucus and debris followed by the use of a second flocked swab to obtain epithelial cells by brushing the mucosa at the level of the inferior turbinate. Venous blood (~2–3 ml) was collected at each visit. RSV infections were confirmed by quantitative reverse transcriptase polymerase chain reaction (qRT-PCR) assay at diagnosis.

RSV disease severity was measured using the Global Respiratory Severity Score (GRSS), calculated using nine weighted clinical variables (general appearance; presence of wheezing, rales, retractions, cyanosis, lethargy, or poor air movement; maximal age-adjusted respiratory rate; and worst room-air oxygen saturation) yielding a score of 1 through 10 [22].

## RNA processing

Four types of RNA-seq data were used in this study: NT (nasal transcriptome), CD4, CD8, and CD19. Technical details for recovering nasal RNA can be found in [30]. Briefly, following flushing of the nares with saline to remove mucus and cellular debris, a flocked swab was used to recover cells at the level of the turbinates. The swab was immediately placed in RNA stabilizer (RNAprotect, Qiagen, Germantown, MD) and stored at 4˚C. Cells were recovered by filtering through a 0.45 uM membrane filter. Recovered cells were lysed and homogenized using the AbsolutelyRNA Miniprep kit (Agilent, Santa Clara, CA) according to the manufacturer's instructions.

CD4, CD8, and CD19 were mRNA expression profiles of the corresponding cell populations purified from peripheral blood as previously described [11,31]. Specifically, heparinized blood was maintained at room temperature for up to 2 hours, and peripheral blood mononuclear cells were isolated by Ficoll-hypaque gradient, flow-sorted into these three subsets of cells, and stored in RNA lysis buffer at −20˚C.

For all four types of RNA-seq data, sequencing libraries were constructed using the NexteraXT library kit (Illumina, San Diego, CA) and then sequenced on the Illumina HiSeq2500 platform. Sequences were aligned against human genome version of hg19 using STARv2.5, counted with HTSeq, and normalized by Fragments Per Kilobase of transcript per Million mapped reads (FPKM). A small subset of samples with very low yields or very low correlation with other samples were removed from the subsequent analyses (Table E in S1 Text), and we applied non-specific filtering based on both mean expression values and inter-quartile range (IQR) to identify subsets of genes for further investigation. We winsorized potential outliers at the gene-level, and then tested the correlation between gene expressions and the GRSS by Pearson correlation test. For each type of transcriptomic data, we were able to select several hundreds of potentially informative features at 0.05 significance level, as summarized in Table F in S1 Text. Additional technical details on data preprocessing can be found in S1 Text.

## Microbiome processing

Bacterial 16 S rRNA from nasal swab specimens was extracted, amplified, and sequenced, and the resulting data were used to determine the taxonomic compositions, in terms of the relative

abundances of those present operational taxonomic units (OTUs). Briefly, the V3-V4 hyper-variable regions were targeted for amplification and sequenced using an Illumina MiSeq platform according to a paired end $2 \times 300$ bp read protocol. Preliminary read processing and quality control were performed using the Quantitative Insights into Microbial Ecology (QIIME) software package [32,33], and a closed-reference OTU picking was done with USEARCH and the GreenGenes reference database [34]. The initial microbiome data contained information for 148 distinct OTUs at the genus-level across 104 samples. Among them, only 15 genera had nonzero abundance level for more than half (n = 52) of the subjects. These features were selected for the integrative analyses. A full list of them is provided in Table G in S1 Text.

## Methods of data integration

We considered five methods of data integration, all of which shared three common components: a uni- or multi-layered dimension reduction to select a manageable set of features from various types of data, and elastic-net regularized regression to integrate these features into one weighted score to predict GRSS, our main outcome variable. Elastic-net regularized regression uses both $L^1$ (LASSO) and $L^2$ (ridge) penalties to produce a sparse linear predictive model and is known to be numerically stable for high-dimensional data. It is implemented by the R package glmnet [35]. Regularization parameters were selected by an initial ten-fold cross-validation. After we obtained a sparse regression model, we re-estimated the linear coefficients by OLS-based procedures to improve the accuracy of modeling fitting. This parameter refinement strategy can improve predictive accuracy and is widely used in high-throughput data analysis [36].

Method 1 performed principal components analysis (PCA) separately on each transcriptomic data set and then uses the resulting PCs together with OTUs representing the nasal microbiota in a penalized regression model of GRSS. Methods 2 performed PCA collectively on all data types and then used the resulting PCs in a penalized regression model of GRSS. Method 3 was identical to Method 1 except for the addition of a second layer of PCA prior to a penalized regression model of GRSS. Method 4 was identical to Method 1 except the nasal microbiota OTUs also undergo dimension reduction via PCA. Finally, Method 5 combined the additions of Methods 3 & 4 resulting in a full two-layer PCA. We assessed the ability of the models produced by each of these methods to predict GRSS through leave-one-out cross-validation.

## Weight assignment and transcription factor analysis

By design, the estimated linear coefficients in our integrative models are weights that represent the importance of principal components, not the features in the original data such as genes and microbes. To enhance the interpretability of these integrative models, we calculated weights for each of the original features based on a backpropagation algorithm. Details of this calculation can be found in "Feature weight calculation" the S1 Text. These weights were then used in the transcription factor analyses. As previously described, conserved binding sites from JASPER across hg19, mm10 and rn6 were identified and were mapped to 2 kb region around Transcription Start Site (TSS) for the transcription factor analysis [37]. A hypergeometric test was performed to identify enriched binding sites [38,39]. Statistical significance was assessed at a p-value threshold of 0.05 without adjustment for multiple testing.

## Pathway analysis

Genes that were identified as significantly correlated with GRSS were subsequently used for canonical pathway identification and upstream regulator analysis using Ingenuity Pathway

Analysis (QIAGEN Silicon Valley, Redwood City, CAQiagen). The combined feature weights were used in pathway analysis to enhance the results and were compared with the results obtained without those weights. Statistical significance was assessed at a p-value threshold of 0.05 without adjustment for multiple testing.

## Supporting information

**S1 Text. Supplemental Materials and Methods and Tables A-G.** The text and tables describe details of the data preprocessing and analysis as well as the demographics of the study cohort. (DOCX)

**S1 Fig. Screeplots for each transcriptomic data set: CD4 cells, CD8 cells, CD19 cells, and nasal epithelial cells.** Vertical lines denote the number of principal components that explain 70% of the variation (dashed) or 80% of the variation (solid). (TIFF)

**S2 Fig. Upset plot showing the number of subjects for which each data modality was obtained, the overlap between these sets, and the distribution of GRSS in each subset.** (TIFF)

**S3 Fig. Biological pathways associated with clinical severity in nasal epithelial cells of the comprehensive model (CM).** Canonical pathways defined by the subset of top 200 (I), 1st quartile (II), 2nd quartile (III) and the set of 993 genes (IV) genes with expression associated with clinical severity were identified using Ingenuity Pathway Analysis (IPA). Shown are the 15 pathways with the lowest significant p-values using Fisher's exact test. (TIFF)

**S4 Fig. Biological pathways associated with clinical severity in CD4 lymphocytes of the comprehensive model (CM).** Canonical pathways defined by the subset of top 200 (I), 1st quartile (II), 2nd quartile (III) and the set of 454 genes (IV) genes with expression associated with clinical severity were identified using Ingenuity Pathway Analysis (IPA). Shown are the 15 pathways with the lowest significant p-values using Fisher's exact test. (TIFF)

**S5 Fig. Regulators associated with clinical severity in nasal epithelial cells of the comprehensive model (CM).** Upstream regulators defined by the subset of top 200 (I), 1st quartile (II), 2nd quartile (III) and the set of 993 genes (IV) genes with expression associated with clinical severity were identified using Ingenuity Pathway Analysis (IPA). Shown are the 15 regulators with the lowest significant p-values using Fisher's exact test. (TIFF)

**S6 Fig. Regulators associated with clinical severity in CD4 lymphocytes of the comprehensive model (CM).** Upstream regulators defined by the subset of top 200 (I), 1st quartile (II), 2nd quartile (III) and the set of 454 genes (IV) genes with expression associated with clinical severity were identified using Ingenuity Pathway Analysis (IPA). Shown are the 15 regulators with the lowest significant p-values. (TIFF)

**S7 Fig. Biological pathways associated with clinical severity in CD4 lymphocytes of the lymphocyte model (LM).** Canonical pathways defined by the subset of top 200 (I), 1st quartile (II), 2nd quartile (III) and the set of 454 genes (IV) genes with expression associated with clinical severity were identified using Ingenuity Pathway Analysis (IPA). Shown are the 15

pathways with the lowest significant p-values using Fisher's exact test.
(TIFF)

**S8 Fig. Biological pathways associated with clinical severity in CD8 cells of the lymphocyte model (LM).** Canonical pathways defined by the subset of top 200 (I), 1st quartile (II), 2nd quartile (III) and the set of 333 genes (IV) genes with expression associated with clinical severity were identified using Ingenuity Pathway Analysis (IPA). Shown are the 15 pathways with the lowest significant p-values using Fisher's exact test.
(TIFF)

**S9 Fig. Biological pathways associated with clinical severity in CD19 cells of the lymphocyte model (LM).** Canonical pathways defined by the subset of top 200 (I), 1st quartile (II), 2nd quartile (III) and the set of 662 genes (IV) genes with expression associated with clinical severity were identified using Ingenuity Pathway Analysis (IPA). Shown are the 15 pathways with the lowest significant p-values using Fisher's exact test.
(TIFF)

**S10 Fig. Regulators associated with clinical severity in CD4 cells of the lymphocyte model (LM).** Upstream regulators defined by the subset of top 200 (I), 1st quartile (II), 2nd quartile (III) and the set of 454 genes (IV) genes with expression associated with clinical severity were identified using Ingenuity Pathway Analysis (IPA). Shown are the 15 regulators with the lowest significant p-values.
(TIFF)

**S11 Fig. Regulators associated with clinical severity in CD8 cells of the lymphocyte model (LM).** Upstream regulators defined by the subset of top 200 (I), 1st quartile (II), 2nd quartile (III) and the set of 333 genes (IV) genes with expression associated with clinical severity were identified using Ingenuity Pathway Analysis (IPA). Shown are the 15 regulators with the lowest significant p-values.
(TIFF)

**S12 Fig. Regulators associated with clinical severity in CD19 cells of the lymphocyte model (LM).** Upstream regulators defined by the subset of top 200 (I), 1st quartile (II), 2nd quartile (III) and the set of 662 genes (IV) genes with expression associated with clinical severity were identified using Ingenuity Pathway Analysis (IPA). Shown are the 15 regulators with the lowest significant p-values.
(TIFF)

**S13 Fig. Biological pathways associated with clinical severity in CD4 cells of an integration model containing CD4 and CD19 lymphocytes.** Canonical pathways defined by the subset of top 200 (I), 1st quartile (II), 2nd quartile (III) and the set of 454 genes (IV) genes with expression associated with clinical severity were identified using Ingenuity Pathway Analysis (IPA). Shown are the 15 pathways with the lowest significant p-values using Fisher's exact test.
(TIFF)

**S14 Fig. Biological pathways associated with clinical severity in CD19 cells of an integration model containing CD4 and CD19 lymphocytes.** Canonical pathways defined by the subset of top 200 (I), 1st quartile (II), 2nd quartile (III) and the set of 662 genes (IV) genes with expression associated with clinical severity were identified using Ingenuity Pathway Analysis (IPA). Shown are the 15 pathways with the lowest significant p-values using Fisher's exact test.
(TIFF)

**S15 Fig. Regulators associated with clinical severity in CD4 cells of an integration model containing CD4 and CD19 lymphocytes.** Upstream regulators defined by the subset of top 200 (I), 1st quartile (II), 2nd quartile (III) and the set of 454 genes (IV) genes with expression associated with clinical severity were identified using Ingenuity Pathway Analysis (IPA). Shown are the 15 regulators with the lowest significant p-values.
(TIFF)

**S16 Fig. Regulators associated with clinical severity in CD19 cells of an integration model containing CD4 and CD19 lymphocytes.** Upstream regulators defined by the subset of top 200 (I), 1st quartile (II), 2nd quartile (III) and the set of 662 genes (IV) genes with expression associated with clinical severity were identified using Ingenuity Pathway Analysis (IPA). Shown are the 15 regulators with the lowest significant p-values.
(TIFF)

**S17 Fig. Biological pathways associated with clinical severity in nasal epithelial cells of an integration model containing NT, CD4, and CD19 cells plus nasal microbiota.** Canonical pathways defined by the subset of top 200 (I), 1st quartile (II), 2nd quartile (III) and the set of 993 genes (IV) genes with expression associated with clinical severity were identified using Ingenuity Pathway Analysis (IPA). Shown are the 15 pathways with the lowest significant p-values using Fisher's exact test.
(TIFF)

**S18 Fig. Biological pathways associated with clinical severity in CD4 lymphocytes of an integration model containing NT, CD4, and CD19 cells plus nasal microbiota.** Canonical pathways defined by the subset of top 200 (I), 1st quartile (II), 2nd quartile (III) and the set of 454 genes (IV) genes with expression associated with clinical severity were identified using Ingenuity Pathway Analysis (IPA). Shown are the 15 pathways with the lowest significant p-values using Fisher's exact test.
(TIFF)

**S19 Fig. Biological pathways associated with clinical severity in CD19 lymphocytes of an integration model containing NT, CD4, and CD19 cells plus nasal microbiota.** Canonical pathways defined by the subset of top 200 (I), 1st quartile (II), 2nd quartile (III) and the set of 662 genes (IV) genes with expression associated with clinical severity were identified using Ingenuity Pathway Analysis (IPA). Shown are the 15 pathways with the lowest significant p-values using Fisher's exact test.
(TIFF)

**S20 Fig. Regulators associated with clinical severity in nasal epithelial cells of an integration model containing NT, CD4, and CD19 cells plus nasal microbiota.** Upstream regulators defined by the subset of top 200 (I), 1st quartile (II), 2nd quartile (III) and the set of 993 genes (IV) genes with expression associated with clinical severity were identified using Ingenuity Pathway Analysis (IPA). Shown are the 15 regulators with the lowest significant p-values using Fisher's exact test.
(TIFF)

**S21 Fig. Regulators associated with clinical severity in CD4 lymphocytes of an integration model containing NT, CD4, and CD19 cells plus nasal microbiota.** Upstream regulators defined by the subset of top 200 (I), 1st quartile (II), 2nd quartile (III) and the set of 454 genes (IV) genes with expression associated with clinical severity were identified using Ingenuity Pathway Analysis (IPA). Shown are the 15 regulators with the lowest significant p-values.
(TIFF)

**S22 Fig. Regulators associated with clinical severity in CD19 lymphocytes of an integration model containing NT, CD4, and CD19 cells plus nasal microbiota.** Upstream regulators defined by the subset of top 200 (I), 1st quartile (II), 2nd quartile (III) and the set of 662 genes (IV) genes with expression associated with clinical severity were identified using Ingenuity Pathway Analysis (IPA). Shown are the 15 regulators with the lowest significant p-values.
(TIFF)

**S23 Fig. Relationship between the total number of mapped reads, enrollment years, and batches in library preparation.** The total number of mapped reads (in millions) is shown on the y-axis stratified by the library preparation batch number; colored boxes denote the enrollment year corresponding to each batch.
(TIFF)

## Acknowledgments

The authors would like to thank the AsPIRES team for critical assistance with subject recruitment and sample collection and the UR Genomic Research Center for processing of the genomic samples. Finally, we are indebted to the patients and families who agreed to participate in these studies.

## Author Contributions

**Conceptualization:** Matthew N. McCall, Mary T. Caserta, Edward E. Walsh, Xing Qiu, Thomas J. Mariani.

**Data curation:** Anthony Corbett, Jeanne Holden-Wiltse.

**Formal analysis:** Matthew N. McCall, Chin-Yi Chu, Lu Wang, Lauren Benoodt, Juilee Thakar, Christopher Slaunwhite, Alex Grier, Xing Qiu, Thomas J. Mariani.

**Funding acquisition:** Ann R. Falsey, David J. Topham, Xing Qiu, Thomas J. Mariani.

**Investigation:** Chin-Yi Chu, Steven R. Gill, Mary T. Caserta, Edward E. Walsh, Thomas J. Mariani.

**Methodology:** Matthew N. McCall, Lu Wang, Lauren Benoodt, Juilee Thakar, Xing Qiu.

**Project administration:** Mary T. Caserta, Edward E. Walsh, Xing Qiu, Thomas J. Mariani.

**Resources:** Ann R. Falsey, David J. Topham, Thomas J. Mariani.

**Software:** Matthew N. McCall, Chin-Yi Chu, Lu Wang, Lauren Benoodt, Juilee Thakar, Xing Qiu.

**Supervision:** Matthew N. McCall, Juilee Thakar, Mary T. Caserta, Edward E. Walsh, Xing Qiu, Thomas J. Mariani.

**Validation:** Matthew N. McCall, Chin-Yi Chu, Lu Wang, Lauren Benoodt, Juilee Thakar, Xing Qiu.

**Visualization:** Matthew N. McCall, Chin-Yi Chu, Lu Wang, Lauren Benoodt, Juilee Thakar, Xing Qiu.

**Writing – original draft:** Matthew N. McCall, Chin-Yi Chu, Lu Wang, Juilee Thakar, Steven R. Gill, Mary T. Caserta, Edward E. Walsh, Xing Qiu, Thomas J. Mariani.

**Writing – review & editing:** Matthew N. McCall, Chin-Yi Chu, Lu Wang, Juilee Thakar, Steven R. Gill, Mary T. Caserta, Edward E. Walsh, Xing Qiu, Thomas J. Mariani.

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
