## [Decision Letter · Decision Letter 0]

21 Jun 2021

Dear Dr. Mariani,

Thank you very much for submitting your manuscript "A systems genomics approach uncovers molecular associates of RSV severity" for consideration at PLOS Computational Biology.

As with all papers reviewed by the journal, your manuscript was reviewed by members of the editorial board and by several independent reviewers. In light of the reviews (below this email), we would like to invite the resubmission of a significantly-revised version that takes into account the reviewers' comments.

We cannot make any decision about publication until we have seen the revised manuscript and your response to the reviewers' comments. Your revised manuscript is also likely to be sent to reviewers for further evaluation.

Sincerely,

Anders Wallqvist

Associate Editor

PLOS Computational Biology

Florian Markowetz

Deputy Editor

PLOS Computational Biology

Reviewer's Responses to Questions

**Comments to the Authors:**

Reviewer #1: 1. METHODS: RNA PROCESSING: Pearson correlation testing of gene expression and GRSS: please specify what kind of normalization and transformation methods are used for correlation. RNA-seq data is not normally distributed.

2. METHODS: PATHWAY ANALYSIS: Combined feature weights used for pathway analysis were compared to the results without those weights – where is this data?

3. RESULTS: 139 infants were enrolled, but supplemental figure 2 represents only 134 subjects

4. RESULTS: Table S1. Subsets of CD4, CD8 and CD19 were sorted from PBMC, what is the range of cell numbers for each population? Why only some lymphocyte RNA-seq data was available for certain infants? Is this because low cell number count for some patients?

5. RESULTS: Figure 2. It would be nice to explain different colors in figure legend.

6. RESULTS: Do you have flow cytometry data of the nasal epithelial cells to show similar cell number/type for each subject? Just to make sure the gene expression difference is not due to the difference in cells present in different cell population.

7. RESULTS: INTEGRATED METHOD DEVELOPMENT: Supplementary Tables 2&3 don’t actually show the comparison between the PCA/elastic net model and other methods for GRSS prediction? Where is the prediction accuracy of method 1 and other methods?

8. RESULTS: Figure 4: what do the colors in the plot for the TFs mean? Please provide color scale key.

9. SUPPLEMENTAL INFO: FEATURE WEIGHT CALCULATION: 2ND PARAGRAPH: Methods 3 and 5 use a secondary PCA, not method 4.

Reviewer #2: Integrative analyses of gene expression data from airway epithelial cells and immune cells with the airway microbiome with the objective to understand host response associated with infant RSV infection severity. The investigators used elastic net regularized regression method, which is a reasonable approach. The findings are interesting and many support prior published literature on the involvement of specific immune pathways and airway microbiome associations. The application of methods to support the integration of high dimensional multi-omics data to understand infant RSV infection severity is novel.

MAJOR COMMENTS:

- Many of the findings support prior individual associations, which while perhaps arguing it decreases novelty, it supports the approach taken.

- A major recognized limitation is the small sample size, but not discussed is the possibly non-random missing data.

- There is no description of cohort participants used in this analysis, specifically age of infants, time from symptom onset to collection of specimens, infant sex, antibiotic administration, and how these covariates were included in the analyses. The infant age, sex, RSV strain and time from symptom onset to collection of biospecimens could well be different based on disease severity and impact the measured host responses. Only a small subset of subjects have all data types, and there is no description of the group of 23 infants with all data types, nor the groups of 61 and 35 subjects, nor the distribution by heathcare or GRSS. It isn’t clear how or if the associations between key covariates and outcomes were included. This is particularly important for sex, infant age at time of infection, RSV strain, and time from symptom onset to sample collection.

- It is also unclear if the described 3 week follow-up biospecimens were used or could be used in these analyses to represent baseline or resolution, or if they were or could be compared with data from acute infection to assess change.

- There should be a discussion of study limitations, as well as a more robust discussion of what supports confidently knowing that L1L1 or L2L2 is the true model.

Minor comments:

- Review articles are cited to support prior findings, and the primary source should be cited in support of specific prior findings

- A number of the individually described findings have been previously published, and these should both be recognized and those manuscripts cited

- Looking at the demographics of the study population in the referenced published methods manuscript, these aren’t “full-term”, but rather “term” infants.

- Methods/”Weight assignment and transcription factor analysis” sub-section. The second to last sentence in this section seems to be missing a key word describing the methods.

Reviewer #3: The manuscript looks original and well organized. Sometimes the reading results not fluent due to the technical details.

Major revisions:

1. The total number of the infant whose RT-PCR confirmed RSV illness is 139. And also, 61 subjects are in the first set and 35 in the second. Do the authors suspect that the number of patients too small? Can this small number of patients be enough to conduct an integrative analysis and deduce general results?

2. The authors used some "preliminary exploratory analyses" to drive the research. Please, can the authors specify which kind of preliminary analyses have been conducted?

3. In the same section ("Results", lines 14-17 ), the authors used some generic words as, "relatively weak", "strong correlation", "weak features". Please, try to support with numerical evidence (tables, percentage, graphs) your assertions.

4. In the same section (Results, line 16), the authors refer to the use of an "appropriate dimension reduction". Please, can you specify which kind of dimension reduction and support your choice?

5. In the same section (Results, line 21), the authors refer to a "shared variation between data types". Please, can you document it?

5. In the section "Model application and interpretation" (line 11-24 and so on), the authors used some generic words as "clearly", "evident", "highest weighted", "unweighted", "high degree", "majority", "significantly activated", etc. Please, can the author try to support these sentences with numerical evidence?

Minor revisions:

1. Supplementary figure 2. It could be better to use a different color for the column with a value of 23

2. Section "Model application and interpretation" (line 20). Please, can the authors better specify which figure they are referring to? It is easy to be lost.

3. Section "Model application and interpretation" (line 28). Please, the authors should insert the reference for the "Ingenuity" application

4. Section "Model application and interpretation" (line 28). The authors mention "Ingenuity suggested that ...": where are the results? In which figure?

5. Section "Model application and interpretation" (line 35). The authors mention "upper right/Panel D": In which figure?

**Have the authors made all data and (if applicable) computational code underlying the findings in their manuscript fully available?**

Reviewer #1: Yes

Reviewer #2: Yes

Reviewer #3: Yes

PLOS authors have the option to publish the peer review history of their article (what does this mean?). If published, this will include your full peer review and any attached files.

Reviewer #1: **Yes: **Zhaohui Xu

Reviewer #2: No

Reviewer #3: **Yes: **Emilio Mastriani, PhD
---

## [Decision Letter · Decision Letter 1]

13 Oct 2021

Dear Dr. Mariani,

Thank you very much for submitting your manuscript "A systems genomics approach uncovers molecular associates of RSV severity" for consideration at PLOS Computational Biology. As with all papers reviewed by the journal, your manuscript was reviewed by members of the editorial board and by several independent reviewers. The reviewers appreciated the attention to an important topic. Based on the reviews, we are likely to accept this manuscript for publication, providing that you modify the manuscript according to the review recommendations.

Sincerely,

Anders Wallqvist

Associate Editor

PLOS Computational Biology

Florian Markowetz

Deputy Editor

PLOS Computational Biology

[LINK]

Reviewer's Responses to Questions

**Comments to the Authors:**

Reviewer #1: it looks good to me

Reviewer #2: The authors have been responsive to the reviewer comments and the manuscript is improved and addresses and intersting question taking a novel and challenging approach. There remains the significant limitation of the sample size and multiple testing. It would be preferable to include a statement to the methods section and the limitations section of the discussion to explain that the findings may be due to type I error due to multiple testing, this may not be obvious to all readers and would be preferable to state.

Suggest adding to the methods and limitations something along the lines of:

We applied a significance threshold for association using a a 2-sided p<0.05 without adjustment for multiple testing, such that findings should be considered exploratory.

Reviewer #3: I wish to thank the authors for their professionality. The article appears to be ready for publication and relevant to the scientific community.

**Have the authors made all data and (if applicable) computational code underlying the findings in their manuscript fully available?**

Reviewer #1: Yes

Reviewer #2: Yes

Reviewer #3: Yes

PLOS authors have the option to publish the peer review history of their article (what does this mean?). If published, this will include your full peer review and any attached files.

Reviewer #1: No

Reviewer #2: No

Reviewer #3: **Yes: **Emilio Mastriani

Figure Files:

Data Requirements:

Reproducibility:

References:

---

## [Editor Report · Decision Letter 2]

5 Nov 2021

Dear Dr. Mariani,

We are pleased to inform you that your manuscript 'A systems genomics approach uncovers molecular associates of RSV severity' has been provisionally accepted for publication in PLOS Computational Biology.

Best regards,

Anders Wallqvist

Associate Editor

PLOS Computational Biology

Florian Markowetz

Deputy Editor

PLOS Computational Biology

---

## [Editor Report · Acceptance letter]

14 Dec 2021

PCOMPBIOL-D-21-00565R2 

A systems genomics approach uncovers molecular associates of RSV severity

Dear Dr Mariani,

I am pleased to inform you that your manuscript has been formally accepted for publication in PLOS Computational Biology. Your manuscript is now with our production department and you will be notified of the publication date in due course.

With kind regards,

Zsofia Freund
